# State-wide California 2020 Carbon Dioxide Budget Estimated with OCO-2 and OCO-3 satellite data

Matthew S. Johnson[1*], Sofia D. Hamilton[2], Seongeun Jeong[2], Yu Yan Cui[3], Dien Wu[4], Alex Turner[5], Marc Fischer[2]

[1]Earth Science Division, NASA Ames Research Center, Moffett Field, CA, USA
[2]Energy Analysis and Environmental Impacts Division, Lawrence Berkeley National Laboratory, Berkeley, CA, USA
[3]Independent Researcher, Sacramento, CA, USA.
[4]Division of Geological and Planetary Sciences, California Institute of Technology, Pasadena, CA, USA
[5]Department of Atmospheric Sciences, University of Washington, Seattle, WA, USA

[*]Correspondence: Matthew S. Johnson (matthew.s.johnson@nasa.gov)

**Abstract**

Satellite observations are instrumental in observing spatiotemporal variability in carbon dioxide ($CO_2$) concentrations which can be used to derive fluxes of this greenhouse gas. This study leverages NASA's Orbiting Carbon Observatory-2 and -3 (OCO-2/3) $CO_2$ observations with a Gaussian Process (GP) machine learning inverse model, a Bayesian non-parametric approach well-suited for integrating the unique spatiotemporal characteristics of these satellite observations, to estimate sub-regional $CO_2$ fluxes. Utilizing the GEOS-Chem chemical transport model (CTM) which simulates column-average $CO_2$ concentrations ($XCO_2$) for 2020 in California – a period marked by the Coronavirus disease (COVID-19) pandemic, drought conditions, and significant wildfire activity – we estimated state-wide $CO_2$ emission rates constrained by OCO-2/3. This study developed prior fossil fuel emissions to reflect reduced activities during the COVID-19 pandemic, while net ecosystem exchange (NEE) and fire emissions were derived based on satellite data. GEOS-Chem source-specific $XCO_2$ concentrations for fossil fuels, NEE, fire, and oceanic sources were simulated coincident to OCO-2/3 $XCO_2$ retrievals to estimate state-wide sector-specific and total $CO_2$ emissions. GP inverse model results suggest annual posterior median fossil fuel emissions were consistent with prior estimates (317.8 and 338.4 Tg $CO_2$ yr$^{-1}$, respectively; 95% confidence level) and that posterior NEE fluxes had less carbon uptake compared to prior fluxes (-36.8 vs. -99.2 Tg $CO_2$ yr$^{-1}$, respectively; 95% confidence level). Posterior fire $CO_2$ emissions were estimated to be 68.0 Tg $CO_2$ yr$^{-1}$ which was much lower compared to a priori estimates (103.3 Tg $CO_2$ yr$^{-1}$). The total median annual $CO_2$ emissions for the state of California in 2020 were estimated to be 349.6 Tg $CO_2$ yr$^{-1}$ (range of 272.8 – 428.6 Tg $CO_2$ yr$^{-1}$; 95% confidence level), aligning closely with the prior total estimate of 342.5 Tg $CO_2$ yr$^{-1}$. This study, for the first time, demonstrates that OCO-2/3 $XCO_2$ observations can be assimilated into inverse models to estimate state-wide, source-specific $CO_2$ fluxes on a seasonal- and annual-scale.

**Short Summary.** Satellites, such as NASA's Orbiting Carbon Observatory-2 and -3 (OCO-2/3), retrieve carbon dioxide ($CO_2$) concentrations which provide vital information for estimating surface $CO_2$ emissions. Here we investigate the ability of OCO-2/3 retrievals to constrain $CO_2$ emissions for the state of California for the major emission sectors (i.e., fossil fuels, net ecosystem exchange, wildfire).

## 1. Introduction

Carbon dioxide ($CO_2$) is the most abundant greenhouse gas in Earth's atmosphere and contributes predominantly to the present-day increase in global radiative forcing (Dunn et al., 2022). Due primarily to anthropogenic emissions from fossil fuel production and usage, global concentrations of $CO_2$ have nearly doubled since the beginning of the pre-industrial era (Gulev et al., 2021; Lan et al., 2023). A recent comprehensive budget analysis of global $CO_2$ fluxes suggests that as of 2022 anthropogenic emissions are ~10 Gt C $yr^{-1}$ (primarily from combustion of coal, oil, and natural gas) with an oceanic and terrestrial uptake offset of ~3 Gt C $yr^{-1}$ and ~4 Gt C $yr^{-1}$, respectively (Friedlingstein et al., 2023). According to this report, the United States (US) contributes 14% (~1.4 Gt C $yr^{-1}$) of global $CO_2$ anthropogenic emissions. The sectors contributing the most to US anthropogenic emissions are transportation, electricity generation, and industry (EPA, 2023). One of the larger emitters of greenhouse gases in the US is the state of California which as of 2021 contributes ~0.1 Gt C $yr^{-1}$ of $CO_2$ (CARB, 2023). In 2006, the state of California passed Assembly Bill 32 (AB 32) which required that by 2020 the state's greenhouse gas emissions must be reduced to 1990 levels. California was able to achieve this goal and in order to demonstrate this, and the success of other future emission reduction goals, it is vital to have accurate estimates of past- and present-day greenhouse gas emissions.

Bottom-up inventories of $CO_2$ are commonly used to derive country-level to state-wide fossil fuel anthropogenic emissions in the US (e.g., Andres et al., 2012; CARB, 2023). Calculations of natural sources and sinks (e.g., terrestrial and marine biosphere, wildfires) contributing to total $CO_2$ emissions are frequently estimated using model predictions (Friedlingstein et al., 2023). The California Air Resources Board (CARB) has quantified state-wide greenhouse gas emissions for California between 2000-2021 (CARB, 2023). Anthropogenic and natural bottom-up $CO_2$ flux estimates are typically implemented in atmospheric transport models and compared to atmospheric observations in order to assess their accuracy. In situ observations of $CO_2$ from ground-based, tower, and aircraft platforms, due to their high accuracy and precision, are most frequently used to evaluate the quality of these emission estimates (Graven et al., 2018; Cui et al., 2022). While highly accurate, these types of in situ observations are limited in their spatiotemporal coverage and ability to constrain large regions and annual cycles of emissions. The assimilation of satellite-retrieved column-averaged dry-air mole fraction of $CO_2$ ($XCO_2$) (e.g., Orbiting Carbon Observatory-2 (OCO-2), Orbiting Carbon Observatory-3 (OCO-3), Greenhouse Gases Observing Satellite (GOSAT), GOSAT-2, Carbon Dioxide Monitoring constellation (CO2M), TanSat) into atmospheric transport models has been demonstrated to be able to constrain emissions on a global- to country-level scale more effectively in regions which lack dense in situ measurement networks (Pandey et al., 2016; Yang et al., 2021; Peiro et al., 2022; Philip et al., 2022; Imasu et al., 2023; Byrne et al., 2023; Noël et al., 2024). This study focuses on $CO_2$ fluxes in the state of California, where OCO-2 and OCO-3 have been used to constrain urban-scale emissions in the mega-city of Los Angeles (Hedelius et al., 2018; Ye et al., 2020; Kiel et al., 2021; Wu et al., 2022; Roten et al., 2023). However, to-date, no studies have demonstrated the capability to evaluate and constrain $CO_2$ fluxes using satellite-retrieved information on a state-wide spatial domain such as California. For California and other states this is important because some state agencies only release state-wide inventories (not specifically for urban areas) and many climate programs are generated based on the state-wide inventories.

While satellite-based atmospheric inverse modeling provides a significantly enhanced method for quantifying $CO_2$ emissions, using satellite observations in atmospheric inversions introduces two principal challenges. These include: 1) incorporating the spatiotemporal covariance inherent in satellite data and 2) accurately estimating the hyperparameters, such as the length scale, of this covariance. Satellite observations contain both spatial and temporal properties, which means that the data have inherent spatial and temporal characteristics that inform us about surface emissions. However, numerous inverse modeling studies have not consistently incorporated both covariance structures (Johnson et al., 2016; Fischer et al., 2017; Cui et al., 2019; Graven et al., 2018; Nathan et al., 2018; Ye et al., 2020; Wu et al., 2022; Roten et al., 2023). While some studies have accounted for both spatial and temporal covariances, they have not determined optimal hyperparameters that align with the satellite observations (e.g., Turner et al., 2020). For example, the length scale parameter is crucial for influencing the covariance, which in turn affects the estimation of the unknown functions; in many cases, however, this parameter is not estimated explicitly for its

optimal value but instead prescribed. In this study, we applied an atmospheric inversion system which fully utilizes
the spatiotemporal properties embedded in satellite data (i.e., OCO-2 and OCO-3). This system is built based on the
Gaussian Process (GP) machine learning (ML) approach enabled by modern Probabilistic Programming Languages
(PPLs). GP is an ML technique that treats predictions as distributions rather than single points, providing a measure
of prediction uncertainty, which is ideal for atmospheric inverse modeling (see Sect. 2.4), because posterior
uncertainties are vital for providing quantitative information on the confidence level of the emissions constraint.
Inversion $CO_2$ models, other than analytical systems, cannot always provide posterior emission uncertainties, and
these estimates can be unreliable and computationally expensive to calculate (e.g., Liu et al., 2014; Bousserez et al.,
2015). The kernels (i.e., covariance functions) of GP models are employed to capture the intricate spatiotemporal
correlation structures of OCO-2/3 data. PPLs have been used in previous studies (e.g., Jeong et al., 2017, 2018), but
modern PPLs provide significantly improved capabilities to implement GP models. Specifically, the built-in functions
for GP kernels in modern PPLs enhance our ability to model the covariance structure of OCO-2/3 data.
This study applies inverse modeling techniques following GP/ML methods described in further detail in Sect.
2.4 to estimate $CO_2$ fluxes in California for a full year of 2020 using $XCO_2$ observations from OCO-2 and OCO-3.
The year 2020 had numerous anomalous features likely impacting total $CO_2$ fluxes in California such as reduced
anthropogenic emissions caused by coronavirus disease pandemic (COVID-19) lockdown procedures (Yañez et al.,
2022), extreme wildfire activity (Jerret et al., 2022; Safford et al., 2022), and drought conditions (Steel et al., 2022).
The impact these types of events have on $CO_2$ fluxes are challenging to predict and difficult to replicate in bottom-up
emission inventories. This study is structured as follows: Sect. 2 presents the forward and inverse models, satellite
observations, and bottom-up emission inventories; Sect. 3 discusses the results of the study, and Sect. 4 contains the
discussion and conclusions.
**2. Methods**
**2.1 GEOS-Chem forward model**
The forward model used to calculate atmospheric concentrations of $CO_2$ corresponding to OCO-2 and OCO-3
observations was the GEOS-Chem (version 14.0.1) chemical transport model (CTM) (Bey et al., 2001; Nassar et al.,
2010). GEOS-Chem was used to simulate $XCO_2$ concentrations corresponding to each OCO-2 and OCO-3 retrieval
for a nested North America domain (10°–70°N, 40°–140°W) driven by Modern-Era Retrospective Analysis for
Research and Applications, Version 2 (MERRA-2) meteorology at 0.5° × 0.625° spatial resolution using 47 vertical
levels from the surface to 0.01 mb. Chemical boundary conditions (BCs) of $CO_2$ used in the nested simulations were
provided by global GEOS-Chem-based 4D-Var data assimilation system runs at 4.0° × 5.0° horizontal spatial
resolution using 47 vertical levels. These global simulations of $CO_2$ for the year 2020 were constrained using inverse
model methods through the assimilation of OCO-2 $XCO_2$ land nadir + land glint (LN+LG) retrievals and global in
situ observations (Philip et al., 2019, 2022). The bottom-up emission inventories for $CO_2$ fluxes from fossil fuel (FF),
net ecosystem exchange (NEE), wildfires, and oceans are described in Sect. 2.2. GEOS-Chem was initialized with
chemical BCs and run for the entire year of 2020 with two months of spin up time.
Total atmospheric $CO_2$ and source-apportioned (i.e., FF, NEE, fire, ocean, and boundary conditions)
concentrations were calculated over California for all OCO-2 and OCO-3 observations. These source-attributed
concentrations were calculated with sensitivity simulations by turning off individual source fluxes or boundary
conditions and comparing these results to the total atmospheric $CO_2$ concentration predictions from simulations with
all sources included. Model-simulated $XCO_2$ corresponding to each OCO-2 and OCO-3 retrieval ($H$) were derived
through the convolution of model $CO_2$ profiles with the column averaging kernel vector ($\boldsymbol{a}$) from OCO-2 and OCO-3
following Eq. (1):
$$H = XCO2_a + \boldsymbol{a}^T(\boldsymbol{f}(\boldsymbol{\sigma}(x)) - \boldsymbol{c_a}) \tag{1}$$
where prior profiles of $CO_2$ ($c_a$) and prior column $CO_2$ ($XCO2_a$) represent prior information used in the OCO-2 and
OCO-3 $XCO_2$ retrieval (O'Dell et al., 2012) and $f(\sigma(x))$ is the GEOS-Chem-predicted vertical profiles of $CO_2$
interpolated to the retrieval levels of OCO-2 and OCO-3.
**2.2 Bottom-up emission inventories**
Bottom-up emission inventories used to drive GEOS-Chem simulations are described in Table 1 and seasonally-
averaged emission maps are displayed in Fig. S1. The Vulcan version 3.0 FF emission inventory covers all
anthropogenic source sectors of $CO_2$ in California (i.e., residential, commercial, industrial, electricity production,
onroad, nonroad, commercial marine vessel, airport, rail, and cement) between 2010-2015 (Gurney et al., 2020). To
create a spatially and temporally resolved Vulcan inventory in California for the year 2020 ($V2020_M$), the hourly 2015
Vulcan emissions ($V2015_M$) are scaled by an annual and a monthly scaling factor using Eq. (2). The sector-specific
annual scaling factor ($R^{CARB}_{2020/2015}$) is calculated as the ratio of annual emissions from that sector in the CARB
inventory for 2020 (which accounts for COVID-19 lockdown emissions reductions; CARB, 2022) to the 2015
emissions. The sector-specific monthly scaling factor ($R_M$) was calculated from activity data from each sector, as the
ratio of monthly activity to annual average activity, and used to appropriately distribute reductions due to the COVID-
19 lockdown throughout the year.
$$V2020_M = V2015_M \times R^{CARB}_{2020/2015} \times R_M \qquad (2)$$
The Vulcan inventory for 2015 was then multiplied by these scaling factors to produce $V2020_M$. Both Vulcan and
CARB provide the same sector-level emission estimates, so the scaling was done for each emission sector separately.
The scaled 2020 Vulcan inventory was then aggregated to $0.1° \times 0.1°$ latitude and longitude.
Natural $CO_2$ emission source (NEE, wildfire, and ocean) estimates were available for the year 2020 and no
scaling was necessary. Biospheric fluxes of $CO_2$ were derived using monthly 5 km $\times$ 5 km NEE calculations from the
Solar-Induced Fluorescence for Modeling Urban biogenic Fluxes version 1 (SMUrF v1; Wu et al., 2021) model.
SMUrF calculates gross primary production (GPP), respiration ($R_{eco}$), and NEE (= $R_{eco}$ – GPP) fluxes using 1) land
cover type 500 m MODerate resolution Imaging Spectroradiometer (MODIS) data, 2) solar induced florescence (SIF)
from the OCO-2 sensor, 3) above ground biomass at 100 m resolution from GlobBiomass, 4) observed flux
measurements from eddy-covariance towers, and 5) gridded soil and air temperature data products. Wildfire $CO_2$
emissions were implemented using a modified Global Fire Emissions Database version 4 (GFED4) data set (van Wees
et al., 2022). This modified version of GFED4 was produced using MODIS burned area and fire detections data with
a spatial resolution of 500 m. Finally, oceanic $CO_2$ fluxes were derived from CarbonTracker (CT2022; Jacobson et
al., 2023) $1° \times 1°$ output. These CT2022 coarse spatial scale fluxes were interpolated to match the GEOS-Chem model
spatial resolution.
**Table 1. Bottom-up prior $CO_2$ emission inventories and 2020 carbon budget (Tg $CO_2$ yr$^{-1}$) for California.**

| Source | Inventory Name | Spatial Res. | Annual Flux: California | Reference |
|--------|----------------|--------------|-------------------------|-----------|
| FF | Vulcan | 1 km $\times$ 1 km | 338.4 | Gurney et al., 2020 |
| NEE | SMUrF | 5 km $\times$ 5 km | -99.2 | Wu et al., 2021 |
| Fire | GFED (modified) | 500 m $\times$ 500 m | 103.3 | van Wees et al., 2022 |
| Ocean | CarbonTracker (CT2022) | 100 km $\times$ 100 km | N/A | Jacobson et al., 2023 |
| Net | | | 342.5 | |

**2.3 OCO-2 and OCO-3 observations**
NASA has two operational satellites with the spatial resolution and precision necessary to constrain point-source to
regional- and global-scale $CO_2$ emissions (i.e., OCO-2 and OCO-3). OCO-2 was launched in 2014 and is a sun-
synchronous polar orbiting satellite which is in the Afternoon Constellation (A-train) of Earth Observing Satellites
with a local overpass time of ~1:30 pm retrieving $XCO_2$ at 1.3 km × 2.3 km spatial resolution (Crisp et al., 2017).
OCO-3 has been onboard the International Space Station (ISS) since 2019 and has an orbital inclination of 51.6°
providing observations at varying times of the day (Eldering et al., 2019). OCO-3 makes orbital observations;
however, differs from OCO-2 as it has the capability to make snapshot area maps (SAMs) which cover 80 km × 80
km at the native spatial resolution of 1.6 km × 2.2 km. The $XCO_2$ retrieval from OCO-2 and OCO-3 both use the
Atmospheric Carbon Observations from Space (ACOS) algorithm (O'Dell et al., 2018) and this study applied version
11r and version 10.4r of OCO-2 and OCO-3, respectively. Retrievals of $XCO_2$ from LN+LG retrievals modes were
used to compare to GEOS-Chem and estimate posterior state-wide $CO_2$ emissions. Since individual high
spatiotemporal OCO-2 and OCO-3 retrievals do not provide independent pieces of information, in this study they are
averaged to the 0.5° × 0.625° spatial resolution of GEOS-Chem. In total, 1614 co-located model-satellite data points
were available during 2020 to evaluate prior $XCO_2$ predictions and constrain posterior $CO_2$ emissions. The seasonal
distribution (meteorological seasons: winter [December, January, February; DJF], spring [March, April, May; MAM],
summer [June, July, August; JJA], fall [September, October, November; SON],) of these co-locations were: 386, 299,
551, and 378 for winter, spring, summer, and fall months, respectively. The spatial distribution of the observational
coverage provided by OCO-2+OCO-3 during 2020 is displayed in Fig. S2.
**2.4 Inverse model technique**
The inverse model developed for this study used a GP/ML framework. GP is a flexible, non-parametric approach,
distinguished by its use of hyperparameters, which defines a prior probability distribution over functions (Williams
and Rasmussen, 2006; Biship, 2007; Murphy, 2022). A GP is fully characterized by its mean function $m(\mathbf{x})$ and kernel
$k(\mathbf{x}, \mathbf{x}')$:

$f(\mathbf{x}) \sim GP\left(m(\mathbf{x}), k(\mathbf{x}, \mathbf{x}')\right)$           (3)

$\boldsymbol{y} = f(\mathbf{x}) + \epsilon$           (4)

where $\boldsymbol{y}$ is the OCO-2 and OCO-3 satellite observation vector, including additive noise $\epsilon$ (i.e., noisy version of $f(\mathbf{x})$).
The noise term ($\epsilon$) is modeled as $\epsilon \sim N(0, \sigma_{noise}^2 \boldsymbol{I})$, where $\boldsymbol{I}$ is the identity matrix and $\sigma_{noise}^2$ is the noise variance
hyperparameter. As described in Text S3, $\sigma_{noise}$ is assigned a Half-Cauchy prior distribution, and its posterior is
inferred using the No-U-Turn Sampler (NUTS) sampler (Hoffman and Gelman, 2014). Although sampling every
possible value of the function $\boldsymbol{f}(\mathbf{x})$ across a continuous domain is supported, we sample a finite set of points (i.e.,
OCO-2/3 observation time and locations), leading to a vector of function values, $\boldsymbol{f} = [f(\mathbf{x}_1), f(\mathbf{x}_2), ..., f(\mathbf{x}_N)]$,
which follows a joint Gaussian distribution with mean vector $\boldsymbol{\mu} = m(\mathbf{x}_1), m(\mathbf{x}_2), ..., m(\mathbf{x}_N)$ and covariance matrix
$[Cov]_{i,j} = k(\mathbf{x}_i, \mathbf{x}_j)$. In this work, the terms "kernel" and "covariance function" are used synonymously.

For our flux inference application, we define the mean function $m(\mathbf{x})$ as:

$m(\mathbf{x}) = \boldsymbol{K}\boldsymbol{\lambda} + \mathbf{D}$           (5)

where $\boldsymbol{K}$ is the input data, a $n \times k$ matrix, derived from GEOS-Chem model predictions, $\boldsymbol{\lambda}$ is a vector ($k \times 1$) of
scaling factors, which quantify the adjustment required for our prior emissions estimates to be consistent with
observations, and $\mathbf{D}$ is the systematic bias. In this work, we estimate a single value for $\mathbf{D}$ for each month in 2020.
Thus, each element of vector $\mathbf{D}$ is populated with the same value for each month of 2020. We assume that this bias
term captures systematic bias due to instrument error, model transport error, and GEOS-Chem BC errors (Jeong et al.,
2017). We show the probability density function of the estimated bias hyperparameter by month in Fig. S3. The
median values range from –0.99 to 0.71 ppm, depending on the month. As noted, this value reflects the combined bias
arising from atmospheric transport, boundary conditions, or other potential sources of error. This approach to
addressing model bias has been applied in previous studies (e.g., Jeong et al., 2017). In this work, we included the
bias term in the mean function [Eq. (5)]. As in Jeong et al. (2017), we model the bias term ($\mathbf{D}$) as a single component
in the GP mean function due to the lack of prior information needed to separate it into identifiable sources (e.g.,

transport or boundary condition errors). Introducing multiple terms without such constraints would risk overfitting and model instability. Here, **D** and λ are considered a GP hyperparameter because it directly scales $m(\mathbf{x})$. This mean function has been widely adopted in atmospheric inverse analysis for estimating greenhouse gas emissions (Jeong et al., 2017; Ye et al., 2020; Ohyama et al., 2023). In GP modeling, it is important to note that the function **Kλ** + **D** is used as the mean of the latent (i.e., unknown) GP function, $f(\mathbf{x})$. In traditional Bayesian inversion methods (e.g., Jeong et al., 2017), the mean function are directly related to **y** in the form **y** = **Kλ** + **D** + ε. The prior distributions for λ and other hyperparameters are described in Text S3.

The second component of a GP is the covariance function (i.e., GP kernel), which dictates how function values at different points relate. For the spatial part of the kernel, we employ the Matérn 5/2 kernel, a widely used covariance function for modeling spatial data (Bevilacqua et al., 2022). The Matérn 5/2 kernel between two spatial points can be expressed as:

$$k(\mathrm{x},\mathrm{x}') = \left(1 + \frac{\sqrt{5}r}{\ell_s} + \frac{5r^2}{3\ell_s^2}\right)\exp\left(-\frac{\sqrt{5}r}{\ell_s}\right) \tag{6}$$

$$r = \sqrt{(x_1 - x'_1)^2 + (x_2 - x'_2)^2} \tag{7}$$

where $r$ is the Euclidean distance between the points x and x′, $x_1$ and $x_2$ represent longitude and latitude, and $\ell_s$ is the spatial length scale. The length scale is typically prescribed, estimated, or computed based on independent data (Baker et al., 2022). In this work, we estimate it simultaneously with other hyperparameters (e.g., the scaling factors). We used the Squared Exponential kernel for the temporal covariance to express the relationship between two temporal points:

$$k_{\mathrm{t}}(\mathrm{x},\mathrm{x}') = \exp\left(-\frac{(x_3 - x'_3)^2}{2\ell_t^2}\right) \tag{8}$$

where $x_3$ denotes the time and $\ell_t$ is the temporal length scale. The spatiotemporal kernel matrix is then constructed by multiplying the spatial and temporal kernels:

$$k_{\mathrm{st}}(\mathbf{x},\mathbf{x}') = \sigma^2 k_{\mathrm{s}}(\mathbf{x},\mathbf{x}') \cdot k_{\mathrm{t}}(\mathbf{x},\mathbf{x}') \tag{9}$$

where $\sigma^2$ denotes the variance of the kernel, which scales the amplitude of the function values predicted by the GP. The spatiotemporal kernel, $k_{\mathrm{st}}$, is realized by element-wise multiplication of the spatial, $k_{\mathrm{s}}$, and temporal, $k_{\mathrm{t}}$, kernels. The resulting spatiotemporal kernel maintains the dimensionality of its constituent kernels.

We perform inversions using three distinct mean functions as depicted in Eq. (5). Model 1 incorporates a systematic bias term **D**, assuming a normal distribution with a mean of 0 and a standard deviation (sd) of 0.5 ppm. Model 2 resembles Model 1 but adopts a standard deviation of 1.0 ppm for the bias term. In contrast, Model 3 excludes the systematic bias term **D** and instead corrects the OCO-2 or OCO-3 a priori and background concentrations by applying scaling factors, thus addressing any biases in the OCO-2 or OCO-3 a priori and background concentrations multiplicatively. We evaluate the three GP models using their expected log pointwise predictive density (ELPD), a metric for model predictive performance. Further details on the model comparison through ELPD are provided in the Supporting Information (Text S1 and Fig. S4).

We employed the Markov chain Monte Carlo (MCMC) method to estimate the hyperparameters of the GP model framework. MCMC has been utilized in several atmospheric inverse modeling studies (Ganesan et al., 2014, 2015; Jeong et al., 2016, 2017, 2018). However, we adopt the NUTS, a modern and advanced MCMC algorithm (Hoffman and Gelman, 2014). We utilized the PyMC PPL (Abril-Pla et al., 2023) to implement the NUTS algorithm for MCMC sampling, generating 4,000 samples each month following a tuning phase of 3,000 steps. More details for the GP model structure and the prior distribution for the hyperparameters are described in Text S2. Using the prior distributions specified in Text S3, we inferred the posterior distributions of the GP hyperparameters using the NUTS algorithm, implemented via the PyMC framework. In our approach, all hyperparameters are estimated jointly. That is, the scaling factors are estimated simultaneously with other parameters, such as the kernel length scales (Jeong et al., 2025). This method enables full posterior inference of the hyperparameters, offering a more robust characterization

of uncertainty than point estimation methods. More details on hyperparameter optimization for the GP model are
provided in Jeong et al. (2025).

**2.5 Evaluation techniques**

Prior and posterior emissions can be indirectly evaluated using atmospheric observations for accuracy and uncertainty
applying normalized mean bias (NMB) and root mean square error (RMSE), respectively (Vermote and Kotchenova,
2008). GEOS-Chem forward and inverse model simulations were evaluated using daily OCO-2 and OCO-3 LN+LG
$XCO_2$ retrievals during 2020. These model predictions were evaluated for each season to determine the accuracy of
prior and posterior emissions and BCs which have large variability throughout the year (see seasonal a priori emissions
in Fig. S1). General statistical parameters were used to evaluate model simulations: NMB, RMSE, correlation
coefficient (R), and simple ordinary least-squares linear regression (slope, y-intercept, etc.). Calculations of NMB are
normalized by OCO-2 and OCO-3 observation values as shown in Eq. (10):
$$NMB = \frac{\sum_{i=1}^{N}(M_i - y_i)}{\sum_{i=1}^{N} y_i}$$
(10)

where $N$ is the total number of model ($M_i$) and OCO-2 and OCO-3 ($y_i$) co-locations. Equation (11) is used to calculate
RMSE values:
$$RMSE = \sqrt{\frac{\sum_{i=1}^{N}(M_i - y_i)^2}{N}}$$
(11)

**3. Results**

**3.1 California prior emissions**

According to prior emission inventories used in this study, the majority of $CO_2$ emitted in California is from
anthropogenic FF sources (see Table 1). The Vulcan FF emission inventory, scaled to 2020 emissions using the CARB
state-wide inventory, suggests that anthropogenic sources contributed 338.4 Tg $CO_2$ yr$^{-1}$, and these sources are
primarily located in the Los Angeles Basin and San Francisco Bay Areas where there are highly populated cities (see
Fig. S1). It is estimated that $CO_2$ emissions in 2020 were reduced by ~10% compared to 2019 due to COVID-19
restrictions (CARB, 2022). According to GFED4, a total of 103.3 Tg $CO_2$ yr$^{-1}$ was emitted from biomass burning
during 2020, which was one of the most active wildfire years in California on record. Figure S1 shows that the majority
of these emissions came from the large wildfires which occurred in northern and central California. These fire
emissions were nearly offset by the biospheric uptake of $CO_2$ in California of -99.2 Tg $CO_2$ yr$^{-1}$ estimated by the
SMUrF model (i.e., our prior model). The largest NEE uptake is estimated to occur in the forested regions of northern
California and the Sierra Nevada Mountains and largest respiration fluxes were in the Sacramento and San Joaquin
Valley areas and the Tulare Basin.
For emission sources other than FF, such as wildfire and NEE, $CO_2$ fluxes in the bottom-up data products
had noticeable seasonality (see Fig. S1). Wildfires in 2020 had pronounced emissions during the summer and fall
months compared to minimal emissions in the winter and spring which is California's rainy season. The fire season
of 2020 was exceptionally active with multiple large complexes occurring between August and September (Keeley
and Syphard, 2021). Prior emissions suggest that in California between August and September fires emitted 95.4 Tg
$CO_2$ which was 92% of the annual total. Biospheric fluxes also displayed large seasonality with largest uptake in the
warmer growing season during the spring and summer and highest respiration rates during the colder months of the
winter and fall. NEE uptake peaked between May and June with average monthly uptake rates of around -27.0 Tg
$CO_2$ while respiration peaked between September and October with average monthly rates ~11.0 Tg $CO_2$. Less
seasonality is apparent in Vulcan 2020 FF emissions for California, with monthly emission rates ranging between 23.0
and 32.0 Tg $CO_2$; however, our CARB-adjusted prior FF model does capture the decrease in anthropogenic $CO_2$
emissions upon the initiation of the COVID-19 lockdown during spring 2020.

## 3.2 Evaluation of model-simulated $XCO_2$ using prior emissions

To indirectly evaluate a priori bottom-up emissions, GEOS-Chem forward model simulations were evaluated with OCO-2 and OCO-3 $XCO_2$ retrievals. Figure 1 shows the comparison of modeled and satellite $XCO_2$ values using prior emissions and observations by season. A timeseries of daily co-located prior and posterior model predicted $XCO_2$ compared to OCO-2/3 observations during the year 2020 is also displayed in Fig. S6 (histogram of annual prior and posterior residuals displayed in Fig. S7). For spring months, GEOS-Chem using prior emissions displayed a slight high bias (NMB=1.1 ppm) and low correlation (R = 0.38) as the model did not capture the variability of $XCO_2$ retrieved by satellites. While the model captures the mean $XCO_2$ values observed, high and low values observed in the spring months were not replicated by the model (linear regression slope = 0.24). A similar evaluation was derived for the winter months as the model had a similar high bias (NMB=1.0 ppm), low correlation (R = 0.39), and relatively low linear regression slope (0.24). A somewhat different comparison was calculated between the model with prior emissions and observations for the summer and fall months. The GEOS-Chem simulations during the summer were able to capture the variability in satellite-retrieved $XCO_2$ values with high correlation (R = 0.73) and linear regression slope of 0.75. The model and prior emissions resulted in a small negative bias during the summer months (NMB=-0.4 ppm). The prior model runs had the least bias in the fall months (NMB=-0.3 ppm) and also displayed moderate correlation (R = 0.52) and linear regression slope (0.34). The evaluation of the prior model displayed similar RMSE values throughout 2020 ranging between 1.4 and 1.8 ppm with the largest random error in the fall months and lowest values in the summer. GEOS-Chem using prior emission displayed biases and errors which varied by season and suggests that observational constraint could improve the estimates of $CO_2$ emission in California. The following sections present the inversion of $CO_2$ emissions when assimilating satellite-derived $XCO_2$ values and the evaluation of posterior emissions.

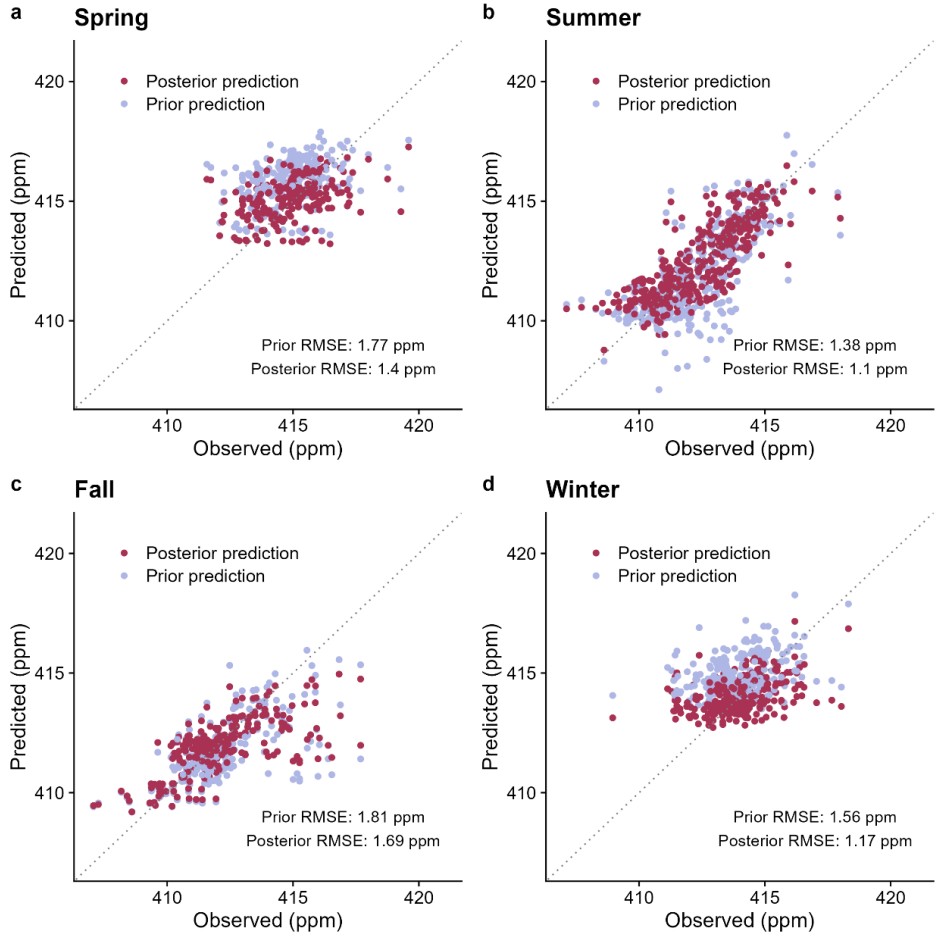

**Figure 1. Comparison of seasonal GEOS-Chem XCO₂ predictions (ppm) using prior (blue) and posterior (purple) emissions and observed OCO-2 and OCO-3 concentrations (ppm). This result is based on the GP inversion model with a systematic bias of 0.5 ppm (Model 1). RMSE values for the prior and posterior model simulations are presented in the figure legend.**

**3.3 Inverse GP model evaluation**

This section describes the evaluation of the GP inversion model using posterior predictive checks (PPCs). PPCs ensure that the inversion results accurately represent the observed data (Gelman et al., 1996). The method involves using the posterior distribution of the model parameters to generate new datasets, which are then compared to the actual observed data. PPCs assess whether the model is capable of producing data similar to the observed data, thereby providing insight into the model's ability to capture the data-generating process accurately. Text S1 describes the comparison of the different GP inversion model setups and how Model 1 performs most accurately. Due to the best performance by Model 1, the rest of the results in this study are based on these outputs. Figure 2 shows PPCs using probability density functions (PDFs) for the middle of each season (except January) using Model 1. Due to an insufficient number of OCO-2 and OCO-3 XCO₂observations (N < 10) in January, the PPC for February is included instead to represent the winter season. We construct the PDFs by utilizing local enhancements in XCO₂ concentrations after subtracting the OCO-2 or OCO-3 a priori XCO₂ and modeled BCs from the total satellite XCO₂ concentrations. The results in Fig. 2 demonstrate that the data generated from the Model 1 posterior parameters generally agree with observations.

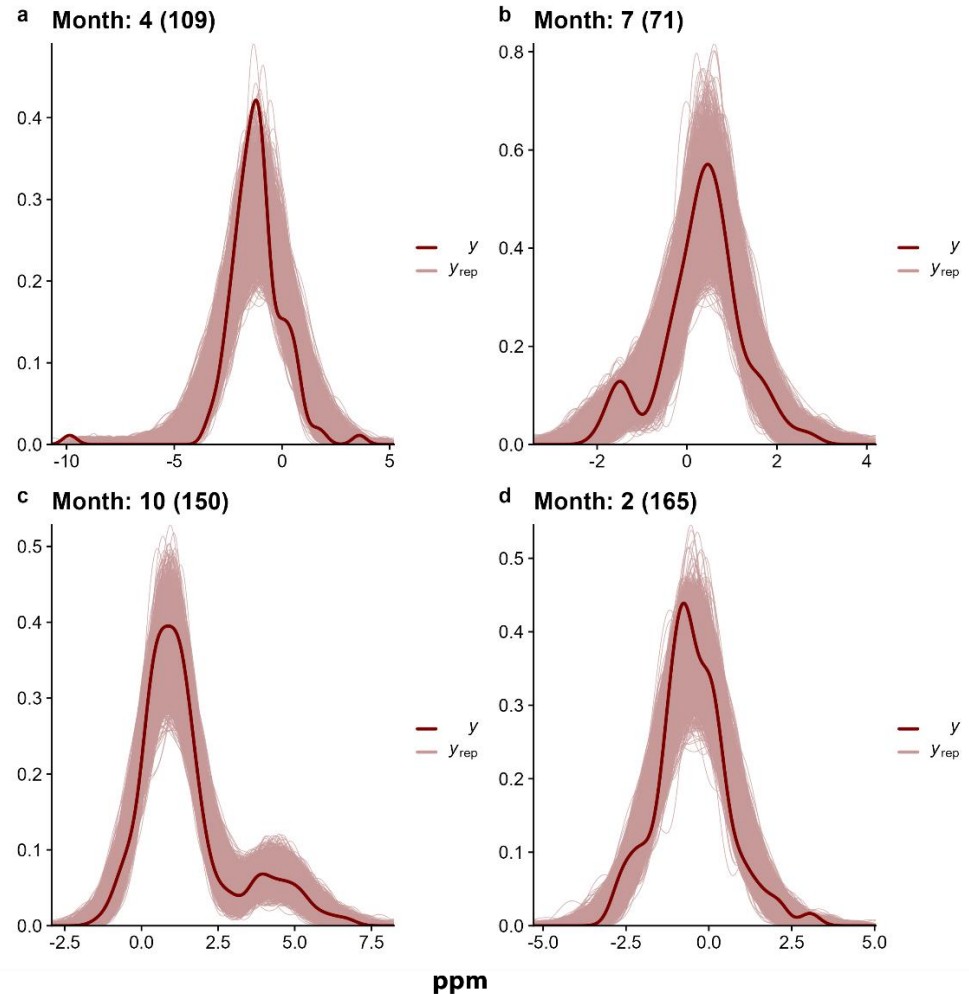

**Figure 2. Evaluation of the GP inverse model (Model 1) performance using PPCs for months representative of each season in 2020 (April = spring; July = summer; October = fall; February = winter). The observed satellite XCO₂ data ($y$; in units of ppm) are represented by the bold lines, while the fine lines ($y_{rep}$) depict 4000 samples (in units of ppm) simulated with parameters drawn from the posterior distributions. Each sample (i.e., each fine line) for each month is of equivalent size to the number of the model-observation co-locations (noted in parentheses).**

The comparison between the posterior predictions from the GP inversion and the observed satellite XCO₂ data indicates an improvement in the RMSE for all seasons (posterior RMSE values on average ~17% lower compared to prior model simulations), suggesting a more accurate model fit than the initial prior predictions (see Fig. 1). The GP inversion was also able to remove the majority of systematic bias imposed by the prior emissions and BCs used in the nested GEOS-Chem simulations along with improving correlation with satellite XCO₂ observations. For spring months, posterior model simulations displayed a small bias of ~0.3 ppm and slightly improved correlation (R = 0.39) compared to prior model results. Posterior model results for the summer season displayed nearly zero bias and high correlation values of 0.79. The statistical evaluation of posterior model performance in the fall months improved compared to prior simulations with bias ~-0.1 ppm and correlation of 0.57. Finally, for winter months posterior results had bias ~0.1 ppm, a significant improvement from 1.0 ppm from the prior result, and moderate correlation of 0.41. Overall, posterior results from the GP inversion performed in this study proved to be more accurate compared to prior simulations suggesting the emission estimates from these inverse model runs are robust, as expected from the PPCs.

## 3.4 Posterior emissions by season and sector

We estimate state-wide posterior emissions by season and sector based on Model 1, which was evaluated to perform the best based on the ELPD metric (see Text S1), and the seasonally-averaged posterior emissions are displayed in Fig. 3. Figure 4 shows the seasonal state-wide total posterior $CO_2$ fluxes from all 3 GP inversion models and the prior estimates for each source sector in California during 2020 (monthly-averaged prior and posterior state-wide $CO_2$ fluxes displayed in Fig. S8). In general, all 3 GP inversion models are relatively consistent with respect to median posterior emissions estimates for all source sectors and seasons. This consistency suggests that the GP models are robust in inferring posterior emissions, despite slight performance variations by season and sector for each model. The rest of the results discussed in this section are focused on posterior estimates from Model 1. Figure 4 shows that posterior FF emissions align closely with the prior estimates on a seasonal-scale, indicating consistency between the initial assumptions and the inversion-derived results. Posterior FF emissions are most consistent with prior estimates during the spring and summer months when COVID-19 lockdown restrictions were most strict, suggesting that corrections applied to the 2020 Vulcan data applying CARB data were reasonable compared to observations. For the fall and winter seasons posterior FF emission estimates were reduced by 10-15 Tg $CO_2$ compared to a priori assumptions, although the reduction is within the margin of error. Seasonal posterior $2\sigma$ uncertainty (95% confidence level) had a range of 20-30 Tg $CO_2$ which on average is ~30% of the seasonal posterior median FF emission values. Interestingly, from the monthly-averaged state-wide emissions shown in Fig. S8, it can be seen that some months in the spring, summer, and fall of 2020 had posterior FF fluxes further reduced compared to the prior emissions emphasizing the strong reduction in GHG emissions due to COVID-19 lockdown restrictions.

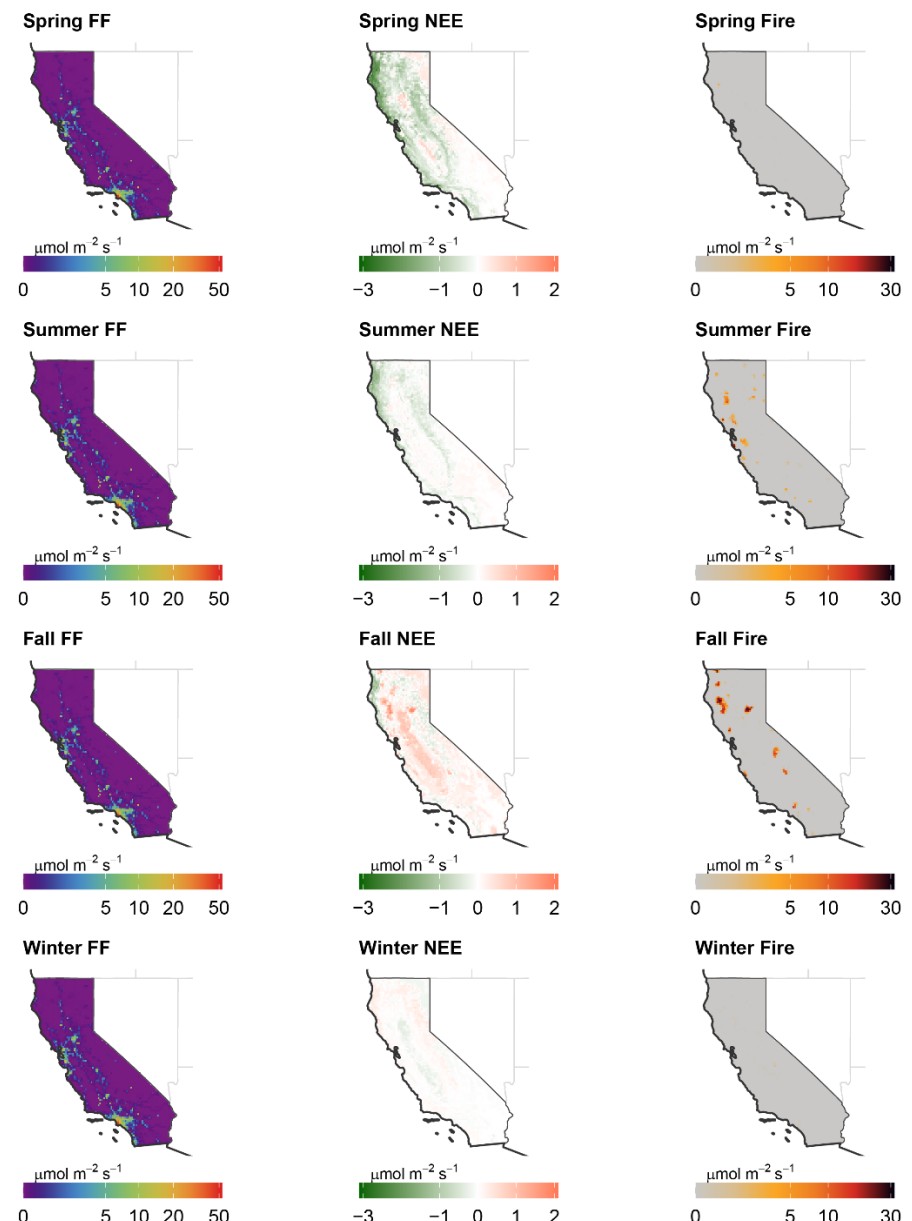

Figure 3. Seasonally-averaged 2020 posterior $CO_2$ emissions (µmol m$^{-2}$ sec$^{-1}$) for the state of California. Emissions from the terrestrial portion of California are shown for FF (left column), NEE (middle column), and Fire (right column) for the spring (first row), summer (second row), fall (third row), and winter (fourth row) months.

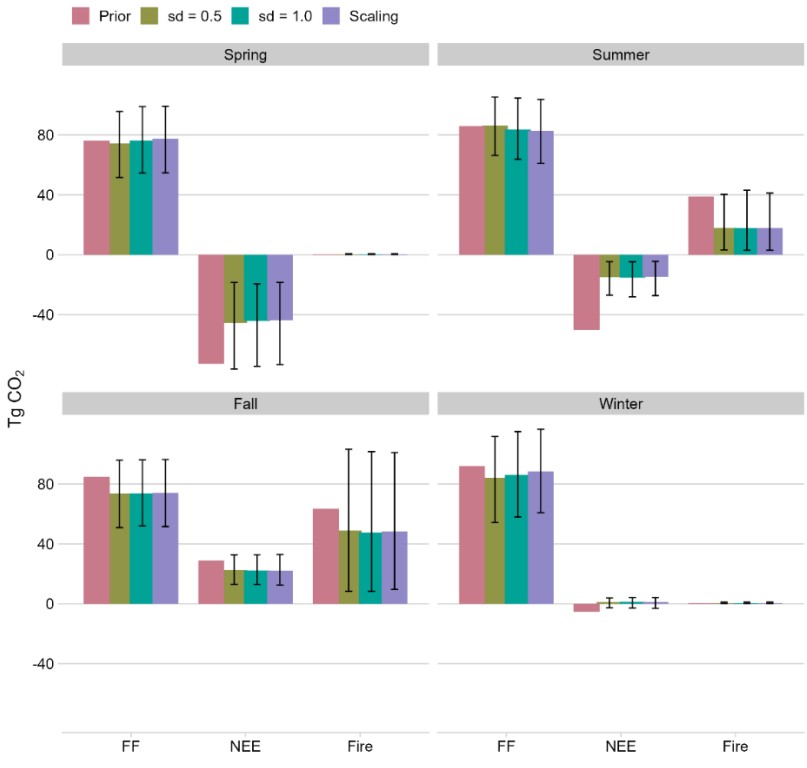

**Figure 4. Sectoral emission estimates (Tg CO$_2$) by season from the scaled Vulcan a priori and those using three distinct models: Model 1 ("sd = 0.5"), where a standard deviation (sd) of 0.5 ppm is applied to the prior probability distribution for the systematic bias; Model 2 ("sd = 1.0"), with a standard deviation of 1.0 ppm for the prior for the systematic bias; and Model 3 ('Scaling'), which optimizes the OCO-2 and OCO-3 a priori and model-predicted BC concentrations using scaling factors analogous to sector emissions adjustments. The error bars in this figure reflect the 2σ uncertainty (i.e., 95% confidence) values for each source sector. All 2σ confidence intervals were calculated using 4000 MCMC samples.**

Posterior NEE fluxes from the GP inversion indicate that prior estimates assumed carbon uptake which was too strong during the drought year of 2020, suggesting an overestimation of the ecosystem's carbon sequestration capacity. Besides the fall season, posterior NEE was much less negative compared to the a priori fluxes, and even transitioned from a small sink to a small source during the winter season (see Fig. 3 and 4). Posterior NEE fluxes are 25–35 Tg CO$_2$ less (lower NEE) compared to prior estimates in the growing seasons of the spring and summer. From Fig. S8 it can be seen that during some of the summer months posterior NEE fluxes were near neutral compared to the large uptake suggested by prior fluxes. This is likely due to the strong drought and hot temperatures experienced in California during 2020 greatly reducing the CO$_2$ uptake during the growing season. The posterior adjustments during the fall were smaller and tended to be consistent with prior estimates from SMUrF. Posterior NEE emissions were consistent with the prior estimates within a 2σ uncertainty range for the spring and fall seasons; however, were not statistically consistent for winter and summer months. Seasonal posterior NEE displayed the largest uncertainty values of all source sectors in California and these uncertainties were on average ~95% of the seasonal posterior median emission value.

The inversion results for fire emissions imply that the prior estimates are consistent with the posterior results within the 2σ uncertainty range although the posterior median values for summer were lower than the prior. As expected, prior and posterior CO$_2$ emissions from fires were small during the winter and spring months. Posterior median seasonal total CO$_2$ emissions ranged between 20 and 50 Tg CO$_2$ for the summer and fall seasons, respectively.

Constraints from OCO-2 and OCO-3 observations reduced emission estimates compared to the prior during both of these seasons with the largest reduction occurring for summer months (-21%). Seasonal posterior fire emissions displayed moderate to high uncertainty values and these uncertainties were on average ~80% of the seasonal posterior median emission values.

**3.5 State-wide posterior total $CO_2$ emissions**

This section describes the annual state-wide $CO_2$ flux estimates constrained using OCO-2 and OCO-3 observations for each source sector in 2020. Table 2 shows the results of the prior and posterior state-wide flux estimates for each source sector and the overall net terrestrial flux. The PDFs of these annual state-wide $CO_2$ fluxes are displayed in Fig. 5 (seasonal sector $CO_2$ flux PDFs shown in Fig. S9). Both the table and figure show that the net state-wide $CO_2$ flux from both prior and posterior estimates are nearly identical between 340-350 Tg $CO_2$ yr$^{-1}$. However, larger differences are evident when the state-wide annual emissions are broken down by source sector. Large constraints were imposed by OCO-2 and OCO-3 observations when focusing on NEE fluxes, where the posterior median estimate (-36.8 Tg $CO_2$ yr$^{-1}$; range of -71.7 – -6.0 Tg $CO_2$ yr$^{-1}$; 95% confidence level) was 63% lower (reduced carbon sink) compared to prior estimates (-99.2 Tg $CO_2$ yr$^{-1}$). Prior emissions from wildland fires were also reduced when constrained by satellite observations as state-wide posterior estimates of 68.0 Tg $CO_2$ yr$^{-1}$ (range of 24.9 – 126.2 Tg $CO_2$ yr$^{-1}$; 95% confidence level) were ~35% lower compared to a priori estimates. Finally, posterior FF emissions were 317.8 Tg $CO_2$ yr$^{-1}$ (range of 271.3 – 364.0 Tg $CO_2$ yr$^{-1}$; 95% confidence level) which is ~5% lower compared to the prior estimates.

**Table 2. Median prior and posterior (2σ range; 95% confidence level) California $CO_2$ budget for 2020.**

| Source | Prior $CO_2$ Flux (Tg $CO_2$ yr$^{-1}$) | Posterior $CO_2$ Flux (Tg $CO_2$ yr$^{-1}$) |
|---|---|---|
| FF | 338.4 | 317.8 (271.3 – 364.0) |
| NEE | -99.2 | -36.8 (71.7 – -6.0) |
| Fire | 103.3 | 68.0 (24.9 – 126.2) |
| Total | 342.5 | 349.6 (272.8 – 428.6) |

For total $CO_2$ fluxes, including all source sectors, California state-wide emissions are constrained with relatively high confidence using OCO-2 and OCO-3 $XCO_2$ observations as the 2σ standard deviation on this total flux is ~23% of the annual median posterior estimate. Annual posterior emission estimates were most confident for FF sources as the 2σ standard deviation from these sources was 47 Tg $CO_2$ which is ~15% of the posterior median value. Natural fluxes of $CO_2$ (i.e., NEE and wildland fire) in California displayed higher uncertainties for their posterior estimates indicated by the wider PDFs in Fig. 5. The 2σ standard deviation of annual posterior NEE fluxes was on average ~35 Tg $CO_2$ which is ~95% of the posterior median value indicating this is the most uncertain carbon flux when using satellites to constrain emissions. Posterior annual fire emissions were also associated with larger uncertainty as the 2σ uncertainty range was 43 Tg $CO_2$ (64% of the median posterior flux).

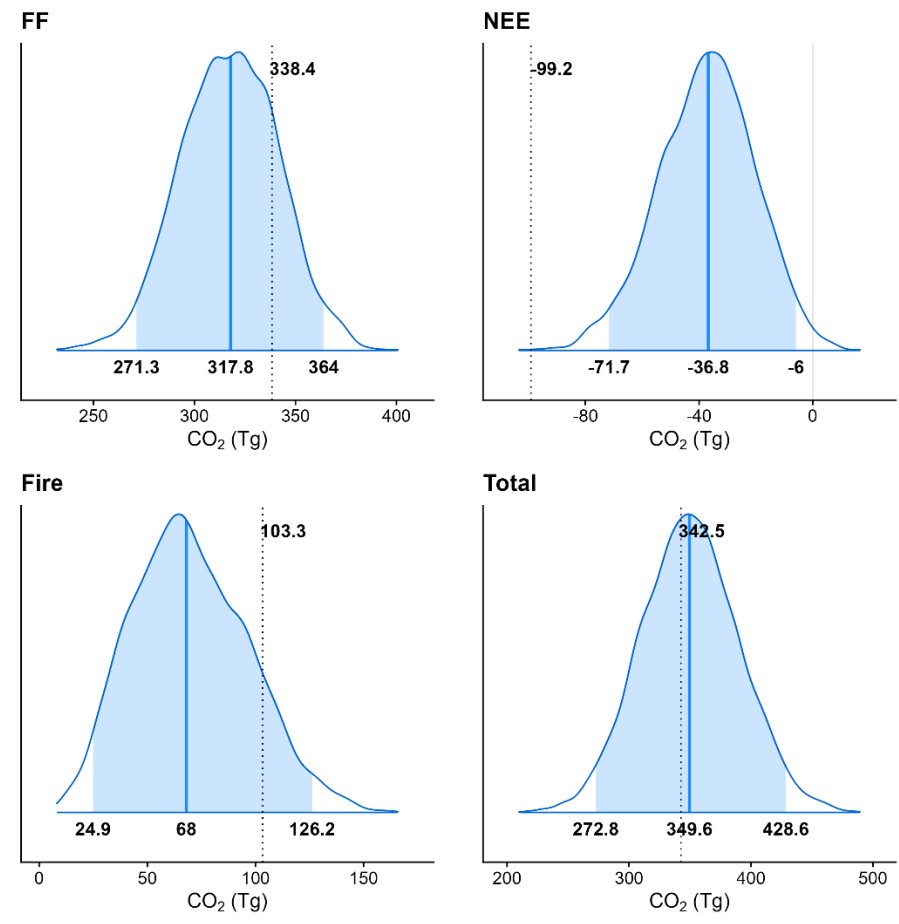

**Figure 5. Annual CO₂ emission totals (Tg CO₂) for California by source sector in 2020. The numerical labels at the base of each PDF denote the 2.5th, 50th (indicated by the bold vertical line), and 97.5th percentile estimates of the posterior emissions, respectively. The vertical dotted line indicates the prior emission estimate, with the corresponding value displayed. Note that the annual PDF presented here was derived by aggregating seasonal MCMC samples.**

**4. Discussion and conclusions**

This study presents the first attempt to constrain state-wide $CO_2$ fluxes from California using spaceborne $XCO_2$ observations using both OCO-2 and OCO-3. We chose to focus on the year 2020 as this time period was characterized by anomalous features likely impacting total $CO_2$ fluxes in California including: reduced anthropogenic emissions caused by the COVID-19 lockdown (Yañez et al., 2022), elevated wildfire activity (Jerret et al., 2022; Safford et al., 2022), and drought occurrence (Steel et al., 2022). Assimilating OCO-2 and OCO-3 LN+LG $XCO_2$ observations into a GP inversion framework was demonstrated in this study to be effective for constraining state-wide $CO_2$ fluxes with a high degree of accuracy. The median posterior top-down annual total $CO_2$ flux of 349.6 Tg $CO_2$ yr$^{-1}$ (range of 272.8 – 428.6 Tg $CO_2$ yr$^{-1}$; 95% confidence level) was consistent with the a priori estimate and constrained with low $2\sigma$ uncertainty levels of ~23%. The posterior uncertainty estimates of this study are similar to other recent research that have used OCO-2 and OCO-3 $XCO_2$ data to constrain city-wide $CO_2$ emissions in California (e.g., Roten et al., 2023), other city flux estimates (e.g., Wu et al., 2020), and country-wide $CO_2$ budgets (e.g., Byrne et al., 2023). Our study adds to the growing evidence of how satellite $XCO_2$ data can be used to confidently estimate city- to country-scale $CO_2$ fluxes.

CARB inventories for the years 2019 and 2020 suggest that anthropogenic FF $CO_2$ emissions were reduced
by ~10% in 2020 compared to the year prior in California. The state-wide annual FF $CO_2$ source was estimated in this
study using the GP inversion assimilating OCO-2 and OCO-3 $XCO_2$ data for 2020 to be 317.8 Tg $CO_2$ yr$^{-1}$ which is
~5% lower than the prior flux assumed. This top-down estimate is ~15% higher compared to the CARB 2020 inventory
which calculated state-wide anthropogenic $CO_2$ emissions for 2020 to be 277.7 Tg $CO_2$ yr$^{-1}$. The state-wide FF $CO_2$
emissions estimated using OCO-2 and OCO-3 data in this study had posterior uncertainties of ~15% on an annual-
scale and therefore is statistically consistent with the CARB 2020 inventory. The difference between the bottom-up
and top-down median FF $CO_2$ emission estimate may be due to errors and uncertainties in the GP inversion and errors
in the bottom-up CARB inventory such as missing sources. It appears the results in our study for FF emission estimates
are robust as they compare well to year 2020 emission totals in California from CARB and posterior top-down
estimates are associated with low posterior uncertainty. Our PPC results, which compare the simulated data from
posterior parameters with observations, provide additional confidence in our GP inversion models.
The main natural sources/sinks of $CO_2$ in California (i.e., NEE and wildland fire) were also estimated in this
study using OCO-2 and OCO-3 $XCO_2$ data. The year 2020 was a year of drought in California and also resulted in
extremely large wildfire activity. The GP inversion resulted in posterior NEE fluxes which were greatly reduced
compared to the initial best guess a priori data. On an annual-scale, the posterior estimate for NEE was -36.8 Tg $CO_2$
yr$^{-1}$ which was 63% lower (reduced carbon sink) compared to prior estimates driven by satellite SIF retrievals. It is
important to note that 2020 was towards the end of a multi-year drought that plagued California and it would be
expected that the terrestrial biosphere would be less effective in its uptake of carbon (Fu et al., 2022). It should also
be noted that the median annual posterior NEE estimates derived in this study with satellite retrievals were associated
with uncertainty levels of ~95%. The larger uncertainty value associated with our posterior NEE estimates, compared
to FF sources, is expected as satellite retrievals are less sensitive to small diffuse signals of $CO_2$ enhancements
associated with the terrestrial biosphere compared to larger FF point-sources. Wildfire activity was elevated in
California during the time of this study, and we estimated that these sources contributed 68.0 Tg $CO_2$ yr$^{-1}$ to the total
state-wide annual carbon budget. The posterior estimate derived in our GP inversion was ~35% lower compared to
prior estimate; however, this estimate still represents highly elevated $CO_2$ emissions from this natural source. CARB
estimated that ~100 Tg $CO_2$ yr$^{-1}$ was emitted from wildfires in 2020 (https://ww2.arb.ca.gov/wildfire-emissions)
which is in line with the prior estimate from GFED4 used in our study. CARB uses an emissions model which is
similar to GFED4 so this is to be expected. The lower posterior wildfire estimate using our GP inversion system was
associated with uncertainty levels (~64% of the median posterior flux) lower compared to NEE and were statistically
consistent with the CARB 2020 state-wide estimate. Prior emission estimates from wildfires are generally uncertain
and satellite observations of the $CO_2$ resulting from these episodic events are challenging; thus, it is not surprising that
posterior fire emissions are one of the more uncertain components of the 2020 California $CO_2$ budget.
Given that individual state- and country-wide $CO_2$ flux data sets generally have over a year of latency,
satellite data becomes vital as this spaceborne data is well equipped to provide more real-time estimates of these
emissions. This is an important aspect of satellite data especially during times of anomalous $CO_2$ fluxes due to
economic activity, wildfire, or flood/drought. Both this study and the work by Roten et al. (2023) clearly demonstrated
the ability of OCO-2, and in particular OCO-3, to help constrain FF emission estimates in California during the
COVID-19 lockdown. OCO-3 is particularly effective for estimating city-wide, or other point- to area-source, fluxes
using the data extracted from SAMs. These area wide observations (~80 km × 80 km) greatly improve the
observational coverage compared to OCO-2 (narrow swath of only ~10 km). These SAMs allow for observations
which reduce errors in assumptions about mixing between the sources and observations and illustrate intra-city
variability of $XCO_2$ which was shown to allow for sector-based emission constraints in California (Roten et al., 2023).
The recent launch of satellites, and future plans for spaceborne instruments, which retrieve greenhouse gas
concentrations (e.g., GHGSat, CO2M, Carbon Mapper, etc.) at high spatial resolution and precision, some of which
will apply SAM observational approaches, should greatly improve the ability to accurately estimate $CO_2$ emissions
from city- to global-scales. As demonstrated, our GP inverse model has the potential to utilize these new satellite data
sets to estimate surface emissions in near-real-time fashion, effectively incorporating the unique spatiotemporal
coverage of space-based information.
In evaluating the GP inversion method used in this study compared to linear inverse classical Bayesian
inversion (CBI) models [e.g., $\mathbf{y}=\mathbf{K}\lambda+\epsilon$, commonly used in atmospheric inversions; (Rodgers, 2000)], advantages and
disadvantages become apparent. The GP-based inversion method employed in this work offers several advantages
over classical methods, as highlighted by Jeong et al. (2025). Jeong et al. (2025) presents the advantages of the GP
method through inversion results from different approaches, and we briefly describe them here, focusing on the key
points.
First, Jeong et al. (2025) demonstrated through simulations using multiple inverse modeling methods for
constraining $CO_2$ fluxes in California when assimilating OCO-2+OCO-3 $XCO_2$ observations that the GP inversion
yields superior results compared to the CBI method. Specifically, the CBI method failed to capture the FF scaling
factor accurately at the 68% confidence level. They repeated the inversion multiple times, and this result was
consistent. Their work also showed that without a proper prior distribution, a simple linear regression produced a
physically implausible scaling factor for fire emissions. In our full Bayesian approach, we specify prior distributions
for all parameters, including the scaling factors and kernel hyperparameters. Overall, the GP inversion offers
substantial flexibility in modeling intricate, non-linear dependencies without needing a pre-specified model
framework (Ebden, 2015).
Second, the CBI method typically relies on analytical solutions and lacks robust techniques for estimating
crucial parameters, such as hyperparameters (e.g., variance of the diagonal elements) for the covariance matrix (i.e.,
the kernel). Consequently, many previous inversion studies have used prescribed values for hyperparameters. For
instance, these studies often utilized known values from other work (e.g., Roten et al., 2023) or estimations derived
from sensitivity analyses to construct the model-data mismatch covariance (Gerbig et al., 2003; Jeong et al., 2013;
Johnson et al., 2016). Such approaches do not guarantee that the estimates are consistent with the observed data. In
contrast, Jeong et al. (2025) showed that the GP method can infer the noise variance, with its median value closely
aligning with the true value. This represents a significant advancement over previous approaches, as it enables the
direct estimation of true noise variance from the input data. Also, the GP inversion intrinsically provides quantification
of uncertainty, which proves advantageous in scenarios with limited data as in atmospheric inversions.
Third, the GP method includes the spatiotemporal kernel as its essential component, as shown in this work.
While some previous work (e.g., Turner et al., 2020) used a covariance with both spatial and temporal components,
many previous inversion studies have not used a fully spatiotemporal covariance. This is because it is not
straightforward to estimate the hyperparameters for the spatiotemporal covariance in the CBI method based on
analytical solutions (Jeong et al., 2025). For example, the work by Turner et al. (2020) did not estimate the covariance
parameters in a way consistent with the data. Incorporating the spatiotemporal covariance as a core component of the
inversion is a significant advantage of the GP method over the CBI method.
While the GP-based inversion has many advantages over CBI methods, the computational demands of the
GP method increase significantly with larger datasets, potentially restricting its application in certain contexts
(Williams and Rasmussen, 2006; Murphy, 2022), although recent development for high-performance computing (e.g.,
GPU-enabled tools) can alleviate this issue. Conversely, the linear inverse model, while less computationally
demanding, assumes linearity and typically requires explicit assumptions about the underlying distribution, which
may not always be valid and can lead to underestimation of model uncertainty (Wang, 2023). Overall, this study
demonstrates the clear advantages of using GP-based inversion techniques and this modeling framework should be
considered for application in future studies for constraining GHG fluxes when assimilating satellite retrievals.
*Code and Data Availability.* The NASA OCO-3 Level 2 bias-corrected version 10.4r and OCO-2 Level 2 bias-
corrected version 11r data are available from https://disc.gsfc.nasa.gov/ (last access: 15 June 2023). The Vulcan
version 3.0 high-resolution hourly dataset is available at https://daac.ornl.gov/cgi-bin/dsviewer.pl?ds_id=1810 (last
access: 27 September 2022). The CARB California GHG Emission inventory is available at
https://ww2.arb.ca.gov/ghg-inventory-data (last access: 2 November 2022). Carbon dioxide fluxes from
CarbonTracker are available from https://gml.noaa.gov/aftp/products/carbontracker/co2/CT-NRT.v2022-
1/fluxes/daily/ (last access: 14 March 2024). Biogenic fluxes from the SMUrF model are available from
http://dienwu.me/gmd2021/ (last access: 29 March 2023). Fire emissions data is available from
https://doi.org/10.5281/zenodo.7229674 (last access: 1 February 2023). The GEOS-Chem model is openly available
to the public and can be downloaded at https://zenodo.org/records/12584192 (last access: 11 September 2023).
*Supplement*. The supplement related to this article is available.
*Author Contributions.* SJ was responsible for obtaining the funding which supported this project. MSJ, SDH, SJ, and
MLF conceived the overall project ideas. MSJ, YC, and SJ performed the GEOS-Chem and GP model simulations
which produced the majority of the results presented in this study. DW, AT, and SDH provided critical bottom-up
emission data sets used throughout the study. MSJ and SJ were responsible for writing the manuscript with the aid of
all coauthors.
*Competing interests.* The authors declare that they have no conflict of interest.
*Acknowledgements.* Computational resources were provided by the NASA High-End Computing Program through the
NASA Advanced Supercomputing Division at NASA Ames Research Center and Computing Allowance Program at
Lawrence Berkeley National Laboratory from the Lawrencium Cluster. The views, opinions and findings of this paper
are those of the authors and should not be construed as an official NASA or United States Government position,
policy, or decision.
*Financial support.* MSJ, SDH, SJ, DW, AT, and MF acknowledge funding support from the NASA Earth Science
Division's Carbon Cycle Science Program (grant number: *80HQTR21T0101*) YC contributions to this work were
through in-kind efforts.

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
