# Peer review of "State-wide California 2020 Carbon Dioxide Budget Estimated 1"

_EGUsphere, 2024_

## Referee Comment (RC2)

**State-wide California 2020 Carbon Dioxide Budget Estimated with OCO-2 and OCO-3 Satellite Data**

By Johnson et al.

This work presented an inversion methodology capable of constraining statewide emissions across California. State of the art datasets and practices were used throughout and results demonstrated success. Total annual posterior estimates for 2020 (the "COVID" year) were in good agreement with prior estimates but certain sector-specific emissions (biospheric and wildfire emissions) demonstrated moderate disagreement. These disagreements led to interesting discussions about drought and fire characteristics during the year 2020. This work highlights the current and future potential of using space-based instruments to constrain CO2 emissions from not only CA, but other states as well. (Of course, global application is also possible, but some input datasets used are only U.S.-covering.) While the paper is overall scientifically sound and well written, there are a few aspects that I feel need some attention. I have arranged my comments such that major notes are listed first (which may require significant attention), followed by minor notes (which may be quickly addressed). For conversational reading/writing, I will refer to all authors as "you". Once these comments are adequately addressed, I recommend this manuscript for publication.

**Major Notes**

1. Apparently, California reduced CO2 emissions to 1990 levels. In **Lines 47-49**, you claimed that *"California was able to achieve this goal but in order to validate this, …, it is vital to have accurate estimates of past- and present-day greenhouse gas emissions."* This is an oddly worded statement since you claim that it was done but imply that it has yet to be validated. (How can you know it was done if it hasn't been validated?) My main complaint here is that the claim of CA reducing emissions to 1990 levels isn't backed up with a source. A curious reader may be interested in digging into how/when this happened. A source should be included here.

2. I am not familiar with all the studies listed in **Lines 75-82**, but this section seems to imply that the length scales in these studies are simply assigned and not estimated. I know that Roten et al., 2023 uses a variogram analysis on *each* SAM to determine the appropriate length scale. Thus, the length scale changes on a per-SAM basis. Do you still consider this to be "prescribed"?

3. What is the Vulcan emission inventory aggregated to? Table 1 lists its native resolution of 1km x 1km; however, the data presented in **Figure 1a** appear to be aggregated. (Perhaps it is merely an illusion?) Nonetheless, I find it odd that the range of flux is 0 to 4 umol/m2/s. This is the same range as NEE (-2 to 2 umol/m2/s)! Considering that similar emission inventories (ODIAC and Hestia) routinely present urban emissions greater that 15 umol/m2/s, the range presented in your figure is suspect. (See the provided figure from Kunik et al.,

2019). If your Vulcan input was in fact aggregated to a coarser resolution, this was not explicitly stated in the text. (Did I miss it somewhere?)

4. To my knowledge, the native temporal resolution of Vulcan is hourly; however, I assume $V_{2015}$ is an annual average? (**Equation 2**). Did you use the hourly Vulcan data in any way? Additionally, I am a bit confused about $C_{2020}$ and $C_{2015}$. Are these monthly as well, such that $(C_{2020})_i$ and $(C_{2015})_i$ where $i$ is a particular month? Given how **Equation 2** is structured, it would make sense that they are annual estimates; however, $C_{2020}$ is *also* used to create monthly scaling factors, $R^{month}$. The way **Lines 132-135** are worded induces some confusion about this. Obviously, if both $C_{2020}$ and $R^{month}$ are monthly data, this would lead to a redundancy. Can this be clarified?

5. How is $\varepsilon$ determined in **Equation 4**? Where does its value(s) come from? I see that it is an error term but I don't think this is really explained within the text.

6. In **Line 338**, I think using "accurate" to describe the corrections applied to Vulcan via CARB is a bit of an overstatement. May I suggest something more conservative like "reasonable"? I say this because, while there is a clear "COVID dip" in the prior emissions during the spring and higher emissions elsewhere (summer, fall, and winter), a similar dip isn't as prominent in the posterior data (sd=0.5ppm). There is a similar dip in the Fall season. In particular, the prior and posterior appear to diverge in the fall, as the posterior decreases while the prior seems to increase. This *could* call into question whether the similar dip in the spring is due to COVID or not. (Perhaps the mechanism decreasing posterior emissions in the Fall is the same thing happening in the Spring?) Furthermore, since the uncertainties are large on posterior estimates, $R^{month}$ values could be significantly increased/decreased while remaining within the posterior error bounds.

7. **Lines 357-359** and **365-367** are oddly worded.

> "*Seasonal posterior NEE values displayed the largest uncertainty values of all source sectors in California and were on average ~95% of the seasonal posterior median emission value.*"

> "*Seasonal posterior fire emission values displayed moderate to high uncertainty values and were on average ~80% of the seasonal posterior median emission values.*"

Do the underlined statements apply to the "red" values or the "purple" values? I think my question is answered by the statement in **Lines 382-383** which says:

> "*Observations as the $2\sigma$ standard deviation on this total flux is ~23% of the annual median posterior estimate.*"

It seems the underlined statements above apply to the uncertainties (purple). Can you make this clearer?

**Minor Updates**

1. Obviously, OCO-2 and OCO-3 are the instruments being used in this study, but there are several other current and future missions that could be highlighted (GOSAT, GOSAT-2, TanSat, CO2M, etc.) (**Lines 60-61**)

2. In **Line 144**, "We et al." should be "Wu et al."

3. Just a suggestion for **Table 1**: include the total annual flux in the table but not in the bottom-line total. (Perhaps include an asterisk beside the value with a note that it isn't CA related?) Some readers may be curious as to what this value is. (This inclusion is certainly optional. Feel free to protest.)

4. The notation for R in **Equation 2** bugs me. For example, $R^{month}$ *could* be seen as $R^3$ for the month of March. Obviously, the context of the equation suggests you wouldn't cube the value of R, but the notation could be clarified a bit.

5. "retrievals" should be "retrieval" in **Line 165**. No "s".

6. From **Line 204**, **Equation 7**: Is the Euclidean distance appropriate? Should it be the Haversine distance instead? Are the points spaced far enough apart for it to matter? *Is this the distance to _aggregated_ OCO-2/3 soundings?*

7. In **Lines 214-215**, **Equation 9**, should this a Kronecker product? Certain Bayesian approaches distribute $k_s$ onto $k_t$ as a Kronecker product instead of an element-wise multiplication. (I'm not implying this is incorrect, I just want to double-check that the appropriate multiplication is, in fact, being used.)

8. Very minor comment: the word "using" appears twice in the same sentence and makes it a little awkward to read (**Lines 232-233**)

9. Another very minor comment: I noticed a pattern of abbreviating carbon dioxide as $CO_2$ and column carbon dioxide as XCO2. Is the inconsistency between the "2" subscript intentional?

10. It would be worth reminding readers that the prior mentioned in **Figure 4** is the *modified* Vulcan product. This could be done in the figure and/or in the caption. In **Lines 411-413**, you mention comparisons to CARB. It may be good to include a CARB estimate in **Figure 4** as well.

11. Can corresponding uncertainties be added to the values in **Table 2**?

---

## Author Comment (AC1)

**Response to Reviewer #2**

We thank the reviewer for their comments and input on the manuscript. We have addressed these comments as described below. All reviewer comments are presented in italic font while the author responses are displayed in standard font. Specific text that was added to the updated manuscript is provided in blue text.

*This work presented an inversion methodology capable of constraining statewide emissions across California. State of the art datasets and practices were used throughout and results demonstrated success. Total annual posterior estimates for 2020 (the "COVID" year) were in good agreement with prior estimates but certain sector-specific emissions (biospheric and wildfire emissions) demonstrated moderate disagreement. These disagreements led to interesting discussions about drought and fire characteristics during the year 2020. This work highlights the current and future potential of using space-based instruments to constrain CO2 emissions from not only CA, but other states as well. (Of course, global application is also possible, but some input datasets used are only U.S.-covering.) While the paper is overall scientifically sound and well written, there are a few aspects that I feel need some attention. I have arranged my comments such that major notes are listed first (which may require significant attention), followed by minor notes (which may be quickly addressed). For conversational reading/writing, I will refer to all authors as "you". Once these comments are adequately addressed, I recommend this manuscript for publication.*

We are happy to hear that the reviewer enjoyed the study and identified the novelty of the methods/results. We have addressed the comments below in order to improve the revised manuscript.

*Major comments:*

*1. Apparently, California reduced CO2 emissions to 1990 levels. In Lines 47-49, you claimed that "California was able to achieve this goal but in order to validate this, ..., it is vital to have accurate estimates of past- and present-day greenhouse gas emissions." This is an oddly worded statement since you claim that it was done but imply that it has yet to be validated. (How can you know it was done if it hasn't been validated?) My main complaint here is that the claim of CA reducing emissions to 1990 levels isn't backed up with a source. A curious reader may be interested in digging into how/when this happened. A source should be included here.*

Thank you to the reviewer for identifying the incorrect phrasing on this sentence. It now reads: "California was able to achieve this goal and in order to demonstrate this, and the success of other future emission reduction goals, it is vital to have accurate estimates of past- and present-day greenhouse gas emissions". This statement was not meant to state the reduction strategy was not validated, but more to state how important it is to quantify state-wide emissions in order to demonstrate the achievement of emission reduction efforts.

*2. I am not familiar with all the studies listed in Lines 75-82, but this section seems to imply that the length scales in these studies are simply assigned and not estimated. I know that Roten et al., 2023 uses a variogram analysis on each SAM to determine the appropriate length scale. Thus, the length scale changes on a per-SAM basis. Do you still consider this to be "prescribed"?*

We thank the reviewer for investigating the details of this statement. First, similar to other studies referenced in our manuscript, Roten et al. (2023) is cited as an example of an approach that does not employ a full spatiotemporal covariance structure. Regarding the use of a length scale estimated from a variogram, although the value is data-driven, it is typically treated as a fixed input to the inverse model. In contrast, our approach jointly infers all parameters—including the length scale—within a full Bayesian framework. Thus, our approach allows their posterior distributions to reflect the observed data, ensuring internal consistency among all model components. However, a variogram-based length scale may not align with the full dataset used in the inversion due to the fact it was estimated separately. More importantly, its uncertainty is not propagated to the posterior distribution of the state vector (i.e., the scaling factors in our case), since it is usually treated as a point estimate.

*3. What is the Vulcan emission inventory aggregated to? Table 1 lists its native resolution of 1km x 1km; however, the data presented in Figure 1a appear to be aggregated. (Perhaps it is merely an illusion?) Nonetheless, I find it odd that the range of flux is 0 to 4 umol/m2/s. This is the same range as NEE (-2 to 2 umol/m2/s)! Considering that similar emission inventories (ODIAC and Hestia) routinely present urban emissions greater that 15 umol/m2/s, the range presented in your figure is suspect. (See the provided figure from Kunik et al., 2019). If your Vulcan input was in fact aggregated to a coarser resolution, this was not explicitly stated in the text. (Did I miss it somewhere?)*

Thank you for identifying this omission in detail. We aggregated the Vulcan inventory to $0.1° \times 0.1°$ latitude and longitude. We now explicitly state this in the revised manuscript. As for the range of emission/flux values displayed, the initial version of Fig. 1 had the upper values saturated at 4 $\mu mol\ m^{-2}\ s^{-1}$. We remade all flux figures (a priori seasonal fluxes are now displayed in Fig. S1 and posterior fluxes in Fig. 3) in the revised manuscript to display maximum emission values up to 50 $\mu mol\ m^{-2}\ s^{-1}$.

*4. To my knowledge, the native temporal resolution of Vulcan is hourly; however, I assume $V_{2015}$ is an annual average? (Equation 2). Did you use the hourly Vulcan data in any way? Additionally, I am a bit confused about $(C_{2020})_i$ and $(C_{2015})_i$. Are these monthly as well, such that $(C_{2020})_i$ and $(C_{2015})_i$ where i is a particular month? Given how Equation 2 is structured, it would make sense that they are annual estimates; however, $C_{2020}$ is also used to create monthly scaling factors, $R^{month}$. The way Lines 132-135 are worded induces some confusion about this. Obviously, if both $C_{2020}$ and $R^{month}$ are monthly data, this would lead to a redundancy. Can this be clarified?*

Thank you for highlighting this confusing wording. We have re-worked the description of the treatment of the Vulcan emissions in Sect. 2.2 of the revised manuscript. Overall, we use the hourly Vulcan emissions and use an annual and monthly scaling factor to both scale the 2015 emissions to the 2020 total, and to redistribute the emissions throughout the year to follow activity patterns affected by COVID-19 lockdowns.

*5. How is $\varepsilon$ determined in Equation 4? Where does its value(s) come from? I see that it is an error term but I don't think this is really explained within the text.*

Thank you for the useful question. For clarification we added the following in the revised manuscript text. "The noise term ($\epsilon$) is modeled as $\epsilon \sim N(0, \sigma_{noise}^2 I)$, where $I$ is the identity matrix and $\sigma_{noise}^2$ is the noise variance hyperparameter. As described in Text S3, $\sigma_{noise}$ is assigned a Half-Cauchy prior distribution, and its posterior is inferred using the No-U-Turn Sampler (NUTS) sampler."

*6. In Line 338, I think using "accurate" to describe the corrections applied to Vulcan via CARB is a bit of an overstatement. May I suggest something more conservative like "reasonable"? I say this because, while there is a clear "COVID dip" in the prior emissions during the spring and higher emissions elsewhere (summer, fall, and winter), a similar dip isn't as prominent in the posterior data (sd=0.5ppm). There is a similar dip in the Fall season. In particular, the prior and posterior appear to diverge in the fall, as the posterior decreases while the prior seems to increase. This could call into question whether the similar dip in the spring is due to COVID or not. (Perhaps the mechanism decreasing posterior emissions in the Fall is the same thing happening in the Spring?) Furthermore, since the uncertainties are large on posterior estimates, $R^{month}$ values could be significantly increased/decreased while remaining within the posterior error bounds.*

We agree with the reviewer for all the above listed reasons and have replaced "accurate" with "reasonable" in the updated manuscript.

*7. Lines 357-359 and 365-367 are oddly worded.*

> *"Seasonal posterior NEE values displayed the largest uncertainty values of all source sectors in California and were on average ~95% of the seasonal posterior median emission value."*

> *"Seasonal posterior fire emission values displayed moderate to high uncertainty values and were on average ~80% of the seasonal posterior median emission values."*

*Do the underlined statements apply to the "red" values or the "purple" values? I think my question is answered by the statement in Lines 382-383 which says:*

> *"Observations as the $2\sigma$ standard deviation on this total flux is ~23% of the annual median posterior estimate."*

*It seems the underlined statements above apply to the uncertainties (purple). Can you make this clearer?*

These sentences were corrected in the updated manuscript:

- Seasonal posterior NEE  displayed the largest uncertainty values of all source sectors in California and these uncertainties were on average ~95% of the seasonal posterior median emission value.
- Seasonal posterior fire emission s displayed moderate to high uncertainty values and these uncertainties were on average ~80% of the seasonal posterior median emission values.

*Minor comments:*

*1. Obviously, OCO-2 and OCO-3 are the instruments being used in this study, but there are several other current and future missions that could be highlighted (GOSAT, GOSAT-2, TanSat, CO2M, etc.) (Lines 60-61)*

I have added acknowledgment and references for these other $CO_2$ observing satellites in the updated manuscript.

*2. In Line 144, "We et al." should be "Wu et al."*

Corrected.

*3. Just a suggestion for Table 1: include the total annual flux in the table but not in the bottom line total. (Perhaps include an asterisk beside the value with a note that it isn't CA related?) Some readers may be curious as to what this value is. (This inclusion is certainly optional. Feel free to protest.)*

The value of 342.5 Tg $CO_2$ $yr^{-1}$ in Table 1 is the net total $CO_2$ flux from California using the a priori flux data. Therefore, we have left this value at the bottom line of the table.

*4. The notation for R in Equation 2 bugs me. For example, $R^{month}$ could be seen as $R^3$ for the month of March. Obviously, the context of the equation suggests you wouldn't cube the value of R, but the notation could be clarified a bit.*

We have reworked the notation of Eq. (2), and the text explaining this equation, in the revised manuscript to avoid this potential confusion. The updated text now reads: "To create a spatially and temporally resolved Vulcan inventory in California for the year 2020 ($V2020_M$), the hourly 2015 Vulcan emissions ($V2015_M$) are scaled by an annual and a monthly scaling factor using Eq. (2). The sector-specific annual scaling factor ($R^{CARB}_{2020/2015}$) is calculated as the ratio of annual emissions from that sector in the CARB inventory for 2020 (which accounts for COVID-19 lockdown emissions reductions; CARB, 2022) to the 2015 emissions. The sector-specific monthly scaling factor ($R_M$) was calculated from activity data from each sector, as the ratio of monthly activity to annual average activity, and used to appropriately distribute reductions due to lockdowns throughout the year.".

The new Eq. (2) now displayed as: $V2020_M = V2015_M \times R^{CARB}_{2020/2015} \times R_M$

*5. "retrievals" should be "retrieval" in Line 165. No "s".*

Corrected.

*6. From Line 204, Equation 7: Is the Euclidean distance appropriate? Should it be the Haversine distance instead? Are the points spaced far enough apart for it to matter? Is this the distance to aggregated OCO-2/3 soundings?*

Thank you for the helpful question. We used the Euclidean distance for convenience, but we note that the Haversine distance would be more accurate. However, because our domain is relatively small and confined to California, the effect of using the Haversine distance would be minimal. We tested the difference between the two using July OCO-2 and OCO-3 $XCO_2$ data, and the difference

in the median value for the fossil fuel sector was less than 2%. We aggregated the observational soundings to the model grid; thus, the distance represents the separation between model grid cells.

*7. In Lines 214-215, Equation 9, should this a Kronecker product? Certain Bayesian approaches distribute $k_s$, onto $k_t$ as a Kronecker product instead of an element-wise multiplication. (I'm not implying this is incorrect, I just want to double-check that the appropriate multiplication is, in fact, being used.).*

Thanks for checking on the specifics of these equations. Here, our input is the full grid of space and time. Therefore, we model the kernel as the product of spatial and temporal kernels, implicitly building a Kronecker structure. We use the full combination of space and time points to take advantage of the built-in functionality in the PyMC probabilistic programming language (PPL).

*8. Very minor comment: the word "using" appears twice in the same sentence and makes it a little awkward to read (Lines 232-233).*

Corrected.

*9. Another very minor comment: I noticed a pattern of abbreviating carbon dioxide as $CO_2$ and column carbon dioxide as XCO2. Is the inconsistency between the "2" subscript intentional?*

This has been corrected throughout the updated manuscript.

*10. It would be worth reminding readers that the prior mentioned in Figure 4 is the modified Vulcan product. This could be done in the figure and/or in the caption. In Lines 411-413, you mention comparisons to CARB. It may be good to include a CARB estimate in Figure 4 as well.*

Good point. The first sentence in the Fig. 4 caption now reads "Sectoral emission estimates (Tg $CO_2$) by season from the scaled Vulcan a priori and those using three distinct models…". To avoid unnecessary confusion about what a priori was applied in our simulations, we decided not to include the CARB estimate in Fig. 4.

*11. Can corresponding uncertainties be added to the values in Table 2?*

The 95% uncertainties have been added to Table 2 in the updated manuscript. The a priori data is not provided with uncertainties, therefore no values have been added for the prior fluxes.

---

## Author Comment (AC2)

**Response to Reviewer #1**

We thank the reviewer for their comments and input on the manuscript. We have addressed these comments as described below. All reviewer comments are presented in italic font while the author responses are displayed in standard font. Specific text that was added to the updated manuscript is provided in blue text.

*The study by Johnson et al. applies a novel inversion technique, i.e., the Gaussian Process machine learning method to infer state-wide, source-specific $CO_2$ fluxes using OCO-2/3 $XCO_2$ data, illustrating that OCO-2/3 $XCO_2$ can be assimilated into inverse models to estimate sub-regional and source-specific $CO_2$ fluxes on a seasonal- and annual-scale. In general, this is a nice study that extended our knowledge on employing new artificial intelligence-based inverse modeling methods to infer $CO_2$ fluxes from atmospheric observations. I have a feeling, the current content looks a bit "thin", some advantages of the new method have not been clearly illustrated. Also, some detailed analyses should be added to make the presented results more sound. I would like to recommend it for publication after addressing the following issues.*

We are glad the reviewer appreciates the novelty of this study and the importance of the inversion method applied. We have addressed the comments below in order to improve the robustness of the analysis.

*Major comments:*

*1. The current results do not clearly illustrate the advantages of the GP/ML inverse method. Comparing the new method with old ones should help.*

We have now improved the discussion of the advantages and disadvantages of the Gaussian Process Machine Learning (GP/ML) inversion system compared to standard atmospheric inverse modeling systems using the following text in the Discussions and conclusion section of the revised manuscript. This specific topic is illustrated in Jeong et al. (2025), that was not yet fully published when this manuscript was submitted for review and thus was not citable, which focuses on comparing the classical Bayesian inversion (CBI) and the GP inversion methods. We have added the following to the text in the revised manuscript:

[revised manuscript text omitted]

*2. The presentation of results:*

- *Figure 2, I expect to see a time-series plot (if it can not be made for pixel scale, for regional scale also work), which can better show the performance of model optimization for prior/posterior and observations. Histogram plots for prior/posterior residuals (simulations minus observations) would also help.*

We agree with the reviewer that this timeseries and histogram information would be helpful for the reader. The figures below were added to the supplemental information and the following text was added to the manuscript: "A timeseries of daily co-located prior and posterior model predicted $XCO_2$ compared to OCO-2/3 observations during the year 2020 is also displayed in Fig. S6 (histogram of annual prior and posterior residuals displayed in Fig. S7)".

[Figure]

Figure S6. Timeseries of daily co-located prior (pink dots) and posterior (blue dots) model predicted $XCO_2$ concentrations (ppm) compared to OCO-2/3 observations (block dots) during the year 2020.

[Figure]

Figure S7. Annually-averaged error residuals for prior (orange) and posterior (blue) simulations (predicted – observed) of column-averaged $XCO_2$ (ppm) in prior and posterior model simulations for the year 2020.

- *Figure S1 showing the spatial distribution of the posterior fluxes in the supplemental files can be moved to the main text.*

Figure S1 shows the seasonally-averaged a priori $CO_2$ emissions used in GEOS-Chem simulations. The caption for this figure has been updated to state this. We have now added a map of the seasonally-averaged posterior emissions to the main body of the revised manuscript (new Fig. 3).

[Figure]

Figure 3. Seasonally-averaged 2020 posterior $CO_2$ emissions (μmol m$^{-2}$ sec$^{-1}$) for the state of California. Emissions from the terrestrial portion of California are shown for FF (left column), Fire (middle column), and NEE (right column) for the spring (first row), summer (second row), fall (third row), and winter (fourth row) months.

- *I expect to see the seasonal cycle (with a monthly time-step) of the prior/posterior $CO_2$ fluxes, for fossil fuels, fires, and NEE. These would help to see if the constrained fluxes can indicate the impact of COVID-19, wildfires, and seasonal anthropogenic emissions. Currently, we only see the results by season.*

The figure below showing the monthly prior and posterior $CO_2$ emissions for California was added to the supplemental information. The addition of this figure resulted in the authors adding a couple

additional findings to the revised manuscript specific to the year 2020 for fossil fuel and NEE $CO_2$ fluxes such as: 1) "Interestingly, from the monthly-averaged state-wide emissions shown in Fig. S8, it can be seen that some months in the spring, summer, and fall of 2020 had posterior fluxes further reduced compared to the prior emissions emphasizing the strong reduction in GHG emissions due to COVID-19 lockdown restrictions" and 2) "From Fig. S8 it can be seen that during some of the summer months posterior NEE fluxes were near neutral compared to the large uptake suggested by prior fluxes. This is likely due to the strong drought and hot temperatures experienced in California during 2020 greatly reducing the $CO_2$ uptake during the growing season".

[Figure]

Figure S8. Monthly prior (red) and posterior (blue) $CO_2$ emissions ($TgCO_2$ yr$^{-1}$) in California from fossil fuel, NEE, and fire fluxes during the year 2020.

*3. Is it possible to perform one-year more inversion? So then we can better understand the performance of the inversion model in revealing the impact of disturbance from COVID-19, wildfires, and droughts.*

We appreciate the reviewer's request to perform additional years of inverse model simulations. However, the authors feel this is outside the scope of this initial work. This specific study focuses on the demonstration of the GP/ML technique for constraining $CO_2$ emissions using satellite retrievals. Due to computational expense, we selected a single year with anomalous emission features in California (i.e., 2020) to demonstrate this technique. Multi-year studies using this inversion technique should however be conducted in future studies.

*4. In Section 2.1, I see the boundary conditions were taken from GEOS-chem 4D-Var run at 4° ×5°, it is relatively coarse compared to the resolution of 0.5° × 0.625° for the inversion. I am not sure if this leads to some uncertainties for the inversion. Some higher-resolution BC, e.g., from CarbonTracker, might be better for the current inversion. Or some tests about the sensitivity of BC can be added.*

Thank you for the comment. We agree that the boundary conditions from a higher resolution product could potentially reduce the uncertainty associated with the boundary conditions. To resolve this potential bias, including the background bias, we use a bias term (**D**) as shown in Eq. (5) of the manuscript. To provide more information about the bias, we added a new figure showing the distribution of the bias hyperparameter for each month in the supplemental information of the revised manuscript. We also added the following text in the revised manuscript [right after Eq. (5)].

"We show the probability density function of the estimated bias hyperparameter by month in Fig. S3. The median values range from –0.99 to 0.71 ppm, depending on the month. As noted, this value reflects the combined bias arising from atmospheric transport, boundary conditions, or other potential sources of error. This approach to addressing model bias has been applied in previous studies (e.g., Jeong et al., 2017). In this work, we included the bias term in the mean function [Eq. (5)]. As in Jeong et al. (2017), we model the bias term (**D**) as a single component in the GP mean function due to the lack of prior information needed to separate it into identifiable sources (e.g., transport or boundary condition errors). Introducing multiple terms without such constraints would risk overfitting and model instability.".

[Figure]

Figure S3. Probability density function for the estimated bias hyperparameter (**D**, ppm) in Eq. (5), shown by month. The shaded area represents the 95% confidence interval, while the vertical line indicates the median value. The bold numbers at the bottom represent the 2.5th, 50th, and 97.5th percentiles.

*Minor comments:*

*1. Line 169, constraining-> constrain?*

This was corrected.

*2. Section 2.3, I expect to see a spatial map showing the data coverage of OCO-2 and OCO-3 XCO$_2$ observations over the study area.*

We thank the reviewer for this comment. We agree that a figure of the observational coverage of OCO-2+OCO-3 would be useful for the reader. The figure below has been implemented into the supplemental information of the revised manuscript.

[Figure]

Figure S2. Seasonally-averaged $XCO_2$ concentrations (ppm) retrieved by OCO-2+OCO-3 during the year 2020 presented at the forward model spatial resolution of $0.5° \times 0.625°$.

*3. How to optimally determine the hyperparameters of the GP/ML model? It is not clear.*

We describe how the hyperparameters were determined in Sect. 2.4 of the original version of the manuscript and provide more details in Text S2 and Text S3 of the original supplemental information document. To improve the description of how this was done, we have now added the following text in Sect. 2.4 of the revised manuscript: "Using the prior distributions specified in Text S3, we inferred the posterior distributions of the GP hyperparameters using the NUTS algorithm, implemented via the PyMC framework. In our approach, all hyperparameters are estimated jointly. That is, the scaling factors are estimated simultaneously with other parameters, such as the kernel length scales (Jeong et al., 2025). This method enables full posterior inference of the hyperparameters, offering a more robust characterization of uncertainty than point estimation methods. More details on hyperparameter optimization for the GP model are provided in Jeong et al. (2025)."